# A 22-G or a 25-G Needle: Which One to Use in the Diagnostics of Solid Pancreatic Lesions? A Systematic Review and Meta-Analysis

**DOI:** 10.3390/cancers16122266

**Published:** 2024-06-19

**Authors:** Łukasz Nawacki, Iwona Gorczyca-Głowacka, Paweł Zieliński, Przemysław Znamirowski, Monika Kozłowska-Geller, Agnieszka Ciba-Stemplewska, Magdalena Kołomańska

**Affiliations:** Collegium Medicum, The Jan Kochanowski University in Kielce, Aleja IX Wieków Kielc 19A, 25-317 Kielce, Polandmagdalena.kolomanska@ujk.edu.pl (M.K.)

**Keywords:** EUS, FNA, biopsy, pancreas, pancreatic cancer, 22-G, 25-G

## Abstract

**Simple Summary:**

Most cases of pancreatic cancer are diagnosed at an advanced stage, where a significant dissemination of the tumor is present. Many patients with locally progressed cancer were subjected to neoadjuvant therapy. To qualify for such treatment, a diagnosis of cancer had to be confirmed. We performed a comparative analysis of two needles of different sizes which are most frequently used in endoscopic ultrasound (EUS)-guided fine needle aspiration (FNA). Fourteen randomized controlled clinical trials were investigated. The outcomes of our meta-analysis revealed that 22-gauge (G) and 25-G needles are of equivalent safety and efficacy as proved by their high accuracy, sensitivity, and specificity in focal pancreatic lesion biopsy acquisition, while not possessing a high risk of biopsy-related complications.

**Abstract:**

With the 12th highest incidence and a common late diagnostic at advanced stages, neoadjuvant therapies for pancreatic cancer are important, but they require a confirmed diagnosis. Being a diagnostic standard, the clarification of the clinical relevance of needle gauges is needed, as larger ones may retrieve more tissue for diagnostics, but may also increase the risk of complications. We performed a meta-analysis to compare the efficiency of the most commonly used 22-G and 25-G needles for EUS guided biopsy in solid pancreatic lesions. The MEDLINE (via PubMed), Embase, Cochrane (CENTRAL), and Scopus databases were searched with “EUS”, “needle”, “FNA”, “pancreas”, “prospective”, “22G”, and “25G” keywords. Mixed effects were assessed in the model, with a mean of 86% and a 95% confidence interval. Fourteen prospective studies that compared the efficiency of 22-G and 25-G biopsy needles in 508 and 524 lesions, respectively, were analyzed, along with 332 specimens biopsied using both needle sizes. The groups did not significantly differ in the outcomes. A low degree of heterogeneity was observed overall, except for specimen adequacy. Moreover, 22-G and 25-G needles have comparable safety and efficacy for focal pancreatic lesion biopsies without a high risk of complications.

## 1. Introduction

Despite the advances in diagnostics and treatment, pancreatic cancer continues to have a high mortality rate, with a 5-year survival rate ranging from 2 to 9%, thus having one of the least favorable prognoses [1]. Moreover, there has been a systematic increase in the number of cases. It is expected that the number of new cases will reach 18.6/100,000 individuals in 2050 (compared with 12.1/100,000 in 2010) [2]. The higher prevalence is closely related to the aging of the population [3], in which the group with the highest risk comprises individuals over 65 years old [4]. Smoking [5], alcohol addiction [6], and, definitely, a family history of pancreatic cancer are well-known risk factors [7]. The current epidemics of obesity is also associated with an increase in the incidence of cancers, such as pancreatic [8]. Furthermore, obesity is linked to cholelithiasis development [9]; thus, it possesses an increased risk of acute pancreatitis, provoking the development of this cancer type in a long-term follow-up in turn [10].

Unfortunately, there are no effective screening tests available for detecting this cancer at an early stage, and most patients present with a locally advanced or metastatic disease. Palliative treatment is the only option for metastases. In a case of locally advanced cancer, neoadjuvant treatment, such as chemotherapy, radiation therapy, or a combination of both, is the preferred choice. Neoadjuvant therapy may allow for the eradication of clinically and radiologically silent metastases and facilitate surgical treatment for locally advanced cancer [11].

While a patient may qualify for surgery without histopathological examinations, a confirmed pancreatic cancer diagnosis is a prerequisite for neoadjuvant treatment in most medical institutions. Two main techniques are available for the acquisition of an affected tissue for pathological examination: percutaneous biopsy under ultrasound or computed tomography (CT) guidance and EUS-guided fine needle biopsy (FNB) that have mostly substituted fine needle aspiration (FNA). The former is considered an option when endoscopic biopsy is not possible [12]. Some papers show that an EUS-guided biopsy or FNA is more effective than other diagnostic methods involving CT [13,14]. The EUS-guided biopsy of focal pancreatic lesions is currently an indispensable diagnostic tool. However, the standard about the performance of the procedure has not been determined to date. Various needle sizes, needle tip types, and collection techniques may differ in their ability to distinguish malignant from benign lesions. This poses a significant risk of misdiagnosis and delayed treatment, which, in the case of pancreatic cancer, can carry serious consequences for collecting material for pathological examination in solid pancreatic lesions.

The most popular needle sizes are 22-G and 25-G [15,16]. In recent years, many meta-analyses have been published evaluating the safety and efficacy of EUS-guided biopsy in the diagnostics of focal pancreatic lesions, focusing on collection techniques [17,18,19] or needle sizes [16,20]; as in the former studies, needle size has been shown to play a substantial role in the higher diagnostic accuracy of the technique than FNA that does not provide sufficient data on tissue architecture.

A 19-G needle can receive a higher amount of material for evaluation than 22-G or 25-G and is the preferred option for histological as opposed to cytological examination, with lower diagnostic accuracy rates than the other two needle sizes. Likewise, a study reported that the overall diagnostic accuracy rate was considerably lower in the 19-G group, where EUS-guided biopsy was carried out, than in the 22-G and 25-G groups, where FNA was performed. The overall accuracy rates for 25-G, 22-G, and Trucut needles were 91.7%, 79.7%, and 54.1%, respectively, and the differences were statistically significant. It was found that the rate of technical successes is greater in 19-G needles and there was a similar accuracy in their use compared to 22-G or 25-G needles. Nonetheless, severe bleeding was a major complication in the 19-G group, while it did not significantly differ in other complications or the rate of technical failures from a higher gauge group [21]. As the mentioned complication is potentially dangerous for a patient, the application of 19-G is restricted. Meanwhile, few data have been reported on the use of 22-G and 25-G needles in EUS-guided biopsies. Moreover, the meta-analyses conducted to date [15] considered both prospective and retrospective longitudinal studies; however, new randomized studies have also emerged and should be included in a new meta-analysis.

Therefore, this study aimed to focus on the safety and efficacy of 22-G and 25-G needles in EUS-guided biopsy and FNA, carry out a comparative analysis of their diagnostic accuracy in solid pancreatic lesions, and clarify the heterogeneities in the outcomes of the previously reported studies.

## 2. Materials and Methods

### 2.1. Search Strategy

The systematic review and meta-analysis were performed according to the recommendations of the Preferred Reporting Items for Systematic Reviews and Meta-Analyses (PRISMA) [22], and the protocol was retrospectively registered under the number INPLASY202450009. The meta-analysis covered only prospective clinical trials comparing 22-G and 25-G needles as used in the biopsy of focal pancreatic lesions by EUS and was conducted using the PICO framework (with Patient, Intervention, Comparison, and Outcomes as the main components) [22,23]. The adequacy and accuracy of the obtained specimen were the primary assessment outcomes, and diagnostic accuracy for malignancy, the number of inadequate biopsies, and complications were the secondary ones. Research works in all languages and publication years were included in the analysis, and pilot studies were not evaluated. Studies where all four parameters (sensitivity, specificity, positive predictive value, and negative predictive value) had a value of 100% were excluded from the evaluation. We adhered to the guidelines for a meta-analysis and made no amendments.

### 2.2. Literature Search

A systematic search was performed on 27 October 2023 in four databases: MEDLINE (via PubMed), Embase, Cochrane (CENTRAL), and Scopus. Two independent review authors (MK and PZ) performed the selection, with an independent senior author (ŁN) responsible for resolving selection disagreements. The keywords were “pancreas”, and “cancer” or “lesion” and “prospective”, and “EUS”, or “needle”, or “biopsy”, or “FNA”, or “22G”, or “25G” in different combinations—two search strings (Appendix A). The titles and abstracts of all retrieved articles were screened. If the titles and abstracts met the indicated inclusion criteria, the full text of an article was analyzed.

### 2.3. Data Extraction and Analysis

Two authors (IGG and PrzZ) independently extracted and validated the information from the selected trials. The following parameters were retrieved: authors’ names, publishing year, type of the study, country, number of cases, specimen adequacy for cytological and histological examinations, specimen accuracy, accuracy for diagnostics of malignancy, the number of inadequate biopsies, the number of complications, sensitivity, and specificity. The Joanna Briggs Institute Critical Appraisal Checklist for Prevalence Studies was used to evaluate the risk of bias in the studies identified [24].

The data from the studies, such as the number of cases with a given endpoint and the determined size of the groups, were necessary for calculating the parameters of the effect of the intervention, which, in this case, was expressed as the number of patients in whom a given event occurred. Intergroup tests were performed to identify statistically significant differences between the results for the two needles (Mann–Whitney U test). For the statistical analysis of continuous data with non-normal distribution and nominal scale, the non-parametric Mann–Whitney U test was used. For categorical variables, Pearson’s chi-squared test of independence was used to delineate the differences between two or more groups of observations, except for the cases involving rare categories with five or fewer observations. In such cases, Fisher’s exact test was used, as recommended. Ultimately, the result of the meta-analysis was the weighted average of the examined effect, graphically presented in the form of a forest plot. Once we conducted the meta-analysis and presented it in the form of a tree chart, it was essential to correctly interpret the data shown on this chart. The vertical main line in it shows no difference between the study groups/interventions (no effect line). Each square represents the results for one study that was covered by the meta-analysis. Its size depends on the weight assigned to it and has a positive correlation with weight. The horizontal lines on each side of a square reflect the size of the confidence interval (usually, we set it at 95%). If at least one line from a square intersected the vertical main line, it meant that the result was not statistically significant. The total outcomes of the analysis of all the included studies are demonstrated in the form of a diamond (rhombus). The wider/more elongated a diamond was, the less precise the confidence interval in which the result fell was, and the narrower a diamond was, the more exact the results of a procedure were. In addition, a list of studies assessed by the meta-analysis is shown on the left side of the chart. When carrying out a meta-analysis, a statistical model with a fixed- or mixed-effect model was used. Assuming that the studies had homogeneous clinical parameters, homogeneity tests were applied to evaluate statistical heterogeneity including the *χ*^2^ test, in which it was assumed that if the *p*-value was below the 0.1 level, the differences between the outcomes of studies were significant, although the results of this test should be treated with caution due to its low power. The second parameter that was taken into account when estimating the homogeneity was the I^2^ parameter, determining the percentage of variation in the estimated effect caused by study heterogeneity and not by a sample selection error. The interpretation of the individual I^2^ ranges that had been suggested in the Cochrane handbook was as follows:0–40%: heterogeneity may not be significant.30–60%: moderate heterogeneity.50–90%: significant heterogeneity.75–100%: very high heterogeneity.

The level of statistical significance α was generally set at 0.05, while there were also 0.01 and 0.001 levels. *p*-values indicating a statistically significant result are highlighted in bold. The relevant notation was always used in the case of *p* < 0.001. In addition, risk difference (RD) and mean difference (MD) coefficients were measured.

Studies where all four parameters (sensitivity, specificity, positive predictive value, and negative predictive value) had a value of 100% were excluded from the analysis; therefore, the results would indicate that the biopsy accurately reflected the gold standard results with no errors without determining the confidence intervals.

## 3. Results

A total of 3246 articles were initially identified by searching the relevant databases, 3232 of which were eliminated during the screening process. A total of 14 articles published between 2006 and 2023 were finally included in the review. Figure 1 shows a PRISMA flow diagram that summarizes the details of the selection process. Table 1 shows the summary of the characteristics of the evaluated studies. A total of 1364 biopsied focal pancreatic lesions were shown in previous reports. Different lesions were biopsied in the comparison between 22-G and 25-G needles in seven studies [25,26,27,28,29,30,31], including 508 and 524 lesions being biopsied using 22-G and 25-G needles, respectively. In the remaining seven studies [15,21,32,33,34,35,36,37], the same focal lesions (332 cases) were diagnosed using both needle sizes.

### 3.1. Specimen Adequacy (Cytology)

To determine whether a 22-G or 25-G needle yielded a better specimen adequacy in EUS-guided FNA biopsies of solid pancreatic lesions, we analyzed the data from eight studies [21,26,29,31,32,33,35,37]. Positive RD values, corresponding to the use of 22-G needles, revealed higher percentages of specimen adequacy (cytology) than the negative RD values which corresponded to the use of 25-G needles.

There were no statistically significant differences in specimen adequacy (cytology) rates between the groups (*p* = 0.136).

There was no significant heterogeneity between the studies (*p* = 0.739); therefore, the abovementioned results were considered as obtained from a fixed model with the exclusion of random factors. The heterogeneity coefficient I^2^ was 0.00% (Figure 2).

Taken together, these findings demonstrate that there was no significant difference in specimen adequacy between the analyzed needle size groups.

### 3.2. Specimen Adequacy (Histology)

Cytology generally involves the exploration of a single cell type, while histological examination covers an entire block of tissue. To determine whether a 22-G or 25-G needle had better specimen adequacy in terms of the latter in EUS-guided FNA biopsies of pancreatic lesions, four studies were encompassed by the meta-analysis [21,30,31,32]. The positive RD values observed in 22-G needles revealed higher rates of specimen adequacy (histology) than the negative RD values observed in 25-G needles.

There were statistically significant differences in specimen adequacy (histology) rates between the groups (*p* = 0.016).

The RD value was 0.2114, showing that specimen adequacy (histology) was 21.14 percentage points higher when using a 22-G needle.

There was no significant heterogeneity between the assessed studies (*p* = 0.026); therefore, the abovementioned results were obtained from a random model. The heterogeneity coefficient I^2^ was 67.52% (Figure 3).

### 3.3. Specimen Accuracy for Diagnostics

To determine whether a 22-G or 25-G needle exhibited better specimen accuracy in EUS-guided FNA biopsies of pancreatic lesions, the data from 12 studies were evaluated in the meta-analysis [21,25,26,28,29,30,31,33,34,35,36,37]. The positive RD values observed in the 22-G needle group correlated with higher rates, and the opposite trend was observed for the negative RD values found in the 25-G needle group.

There were no statistically significant differences in specimen accuracy rates between the groups (*p* = 0.73).

There was no significant heterogeneity between the studies (*p* = 0.55); therefore, the aforementioned results comprised a fixed model. The heterogeneity coefficient I^2^ was 43.19% (Figure 4).

### 3.4. Accuracy for Malignancy

Six studies were analyzed for accuracy for malignancy in the meta-analysis [25,26,28,29,33,36]. The positive RD values corresponding to the 22-G needle group indicated higher rates of accuracy for malignancy than the negative RD values corresponding to the 25-G needle group.

There were no statistically significant differences in accuracy or malignancy rates between the groups (*p* = 0.676).

There was no significant heterogeneity between the studies (*p* = 0.682); therefore, a fixed model was suggested. The heterogeneity coefficient I^2^ was 0.00% (Figure 5).

Appendix A shows the particular number of malignant neoplasia cases.

### 3.5. Inadequate Biopsies

Five studies were included in the evaluation of inadequate biopsies [25,27,30,34,36]. The positive RD values observed in the 22-G needle group had higher rates of specimen adequacy (histology) than the negative RD values observed in the 25-G needle group.

There were statistically significant differences in biopsy rates between the groups (*p* = 0.023).

The RD value was 0.0418, exceeding the percentage of inadequate biopsies by 4.18 percentage points when using a 22-G needle.

There was no significant heterogeneity between the studies (*p* = 0.619); therefore, the aforementioned results belonged to a fixed model. The heterogeneity coefficient I^2^ was 0.00% (Figure 6).

### 3.6. Mean Size of Lesions

Five studies were assessed for a mean size of lesions [25,26,27,28,29], and the MD coefficient was calculated. Positive MD values were higher and were noticed in the 22-G group, and negative values were seen in the 25-G group.

There were no statistically significant differences in the mean size of the lesions between the groups (*p* = 0.706).

There was a significant heterogeneity between the studies (*p* < 0.001); therefore, a random model was proposed. The value of heterogeneity coefficient I^2^ was 91.12% (Figure 7).

### 3.7. Complications

Nine studies were covered by the meta-analysis [26,27,28,29,30,31,33,34,36]. Positive RD values corresponding to higher complication rates were detected in the 22-G needle group, and negative values corresponding to lower complication rates were related to the 25-G needle group.

There were no statistically significant differences in complication rates between the groups (*p* = 0.103).

There was no significant heterogeneity between the studies (*p* = 0.648); therefore, the relevance of a fixed model was confirmed. The value of the heterogeneity coefficient I^2^ was 0.00% (Figure 8).

### 3.8. Sensitivity

Four studies were included for the assessment of sensitivity [30,34,35,36]. Positive RD values stood for higher sensitivity and were obtained with a 22-G needle.

There were no statistically significant differences in sensitivity between the groups (*p* = 0.103).

There was no significant heterogeneity between the studies (*p* = 0.539); therefore, the abovementioned results conformed to a fixed model. The heterogeneity coefficient I^2^ was 0.00% (Figure 9).

## 4. Discussion

Previous meta-analyses reported similar accuracy and number of passes in two needle types [15,20], while we extended the scope of evaluation in our research.

First, there were no statistically significant differences between the mean sizes of lesions being biopsied. Furthermore, there were no significant differences in terms of specimen adequacy for cytology or histology accuracy. In contrast, 25-G needles had a significantly higher adequacy for histological and cytological diagnostics of the lesions in the head and uncinate process with greater accuracy and technical success rates, thus enabling to preserve histological and cytological architecture to a greater extent and elevating the overall quality of the obtained samples [21]. Our results about the adequacy of specimen contradict those of a previous meta-analysis, where 25-G needles were also shown to be superior to 22-G needles [38]. This analysis showed that using a larger diameter needle (22-G vs. 25-G needles) was associated with an increased specimen adequacy for histology compared to using a smaller one; however, it was also linked to a rise in the number of inadequate biopsies. The reason can be the heterogeneity of the assessed samples, not considering the location and size of the tumor, and a greater number of evaluated studies in turn.

Studies comparing 22-G and 25-G sizes generally show no significant differences in diagnostic accuracy, but the use of 22-G is associated with a greater risk of mechanical traumatization while having a lower rate of technical failures [25,32,36].

Theoretically, larger diameter needles (22-G or larger) were supposed to result in more obtained material for histopathology; however, this could be associated with increased traumatization with a subsequently higher number of complications. There were no statistically significant differences in complication rates when using either needle.

The overall diagnostic quality of 22-G needles was also generally compatible to that of 25-G needles in EUS-guided biopsies. This is consistent with the findings of a former single-center study comparing these sizes along with 19-G [39]. However, it was not randomized and could not be included in our meta-analysis. Likewise, there was a considerably smaller number of studies on FNB; hence, the presented amount of data on EUS-guided biopsies in our meta-analysis is limited.

There were no significant differences in sensitivity and specificity between the compared types of needles that can be explained by the greater size of the analyzed sample in our research [40]. Comparing our study with the meta-analysis conducted by Facciorusso et al. [20], we had the opportunity to include new studies to effectively double their number.

This study has some limitations. First, the evaluated studies had a considerable heterogeneity of parameters, potentially interfering with the validity of the findings of the meta-analysis. This can be explained by the combined assessment of both outcomes of EUS-guided biopsy and FNA in the included articles, where the latter prevailed due to a limited number of studies on FNB. Thus, the clinical applicability of the findings is limited in this respect. Similarly, characteristics like tumor size and location, along with other technical aspects of FNA including newly designed needles, were not taken into account. Therefore, a broader study analyzing EUS-guided FNA and FNB separately is warranted for a greater generalizability of the outcomes.

## 5. Conclusions

Based on our meta-analysis, we conclude that the use of a technically simpler-to-use 25-G needle has the same diagnostic quality in cancer as a 22-G needle in EUS-guided biopsies or FNA. Both needles have comparable complication rates, thus being characterized by similar clinical safety and efficacy. Therefore, further studies on other aspects of pancreatic lesion biopsies (types of knives, the use of ROSE technique, etc.) are needed to validate the results.

## Figures and Tables

**Figure 1 cancers-16-02266-f001:**
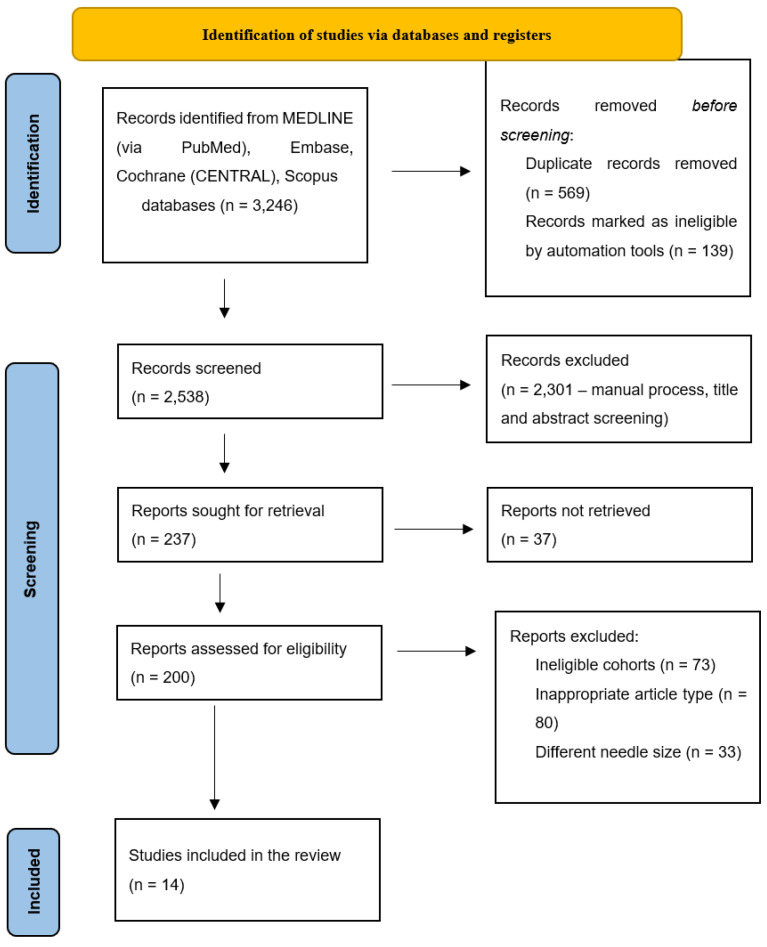
PRISMA flowchart illustrating the literature search and the application of selection criteria.

**Figure 2 cancers-16-02266-f002:**
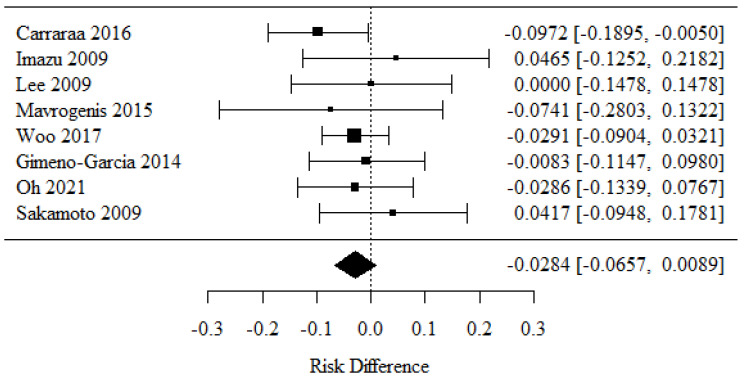
Specimen adequacy in cytology [21,26,29,31,32,33,35,37].

**Figure 3 cancers-16-02266-f003:**
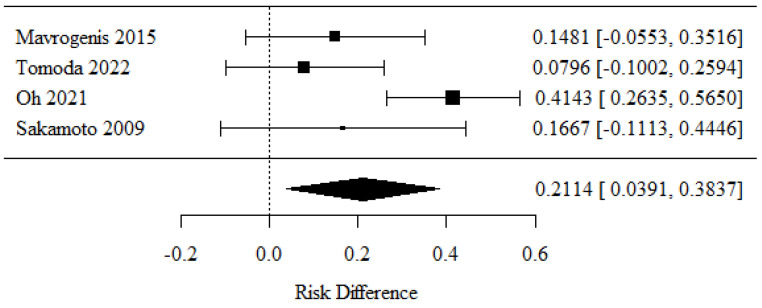
Specimen adequacy in histological examinations [21,30,31,32].

**Figure 4 cancers-16-02266-f004:**
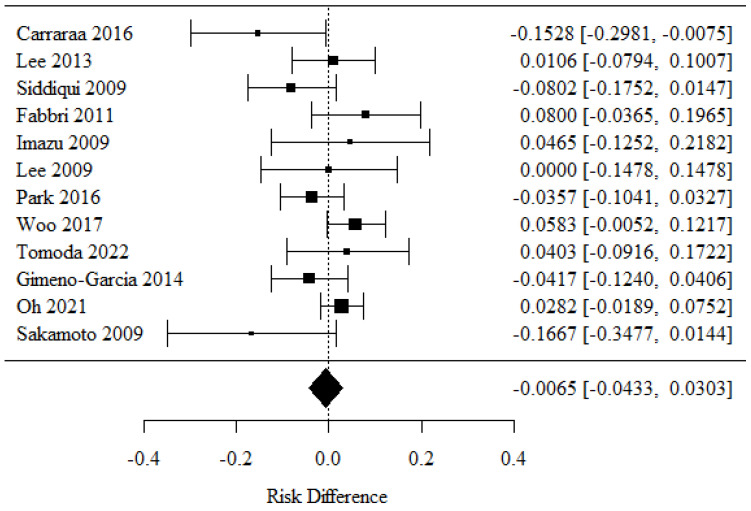
Specimen accuracy for diagnostics [21,25,26,28,29,30,31,33,34,35,36,37].

**Figure 5 cancers-16-02266-f005:**
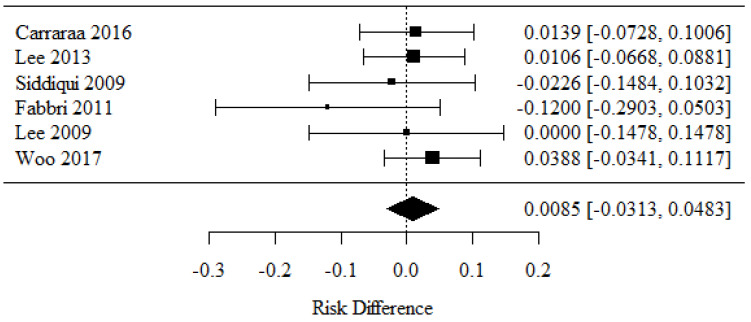
Accuracy for malignancy [25,26,28,29,33,36].

**Figure 6 cancers-16-02266-f006:**
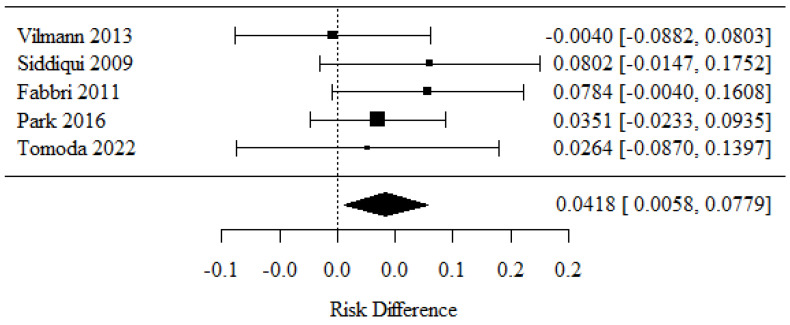
Inadequate biopsies [25,27,30,34,36].

**Figure 7 cancers-16-02266-f007:**
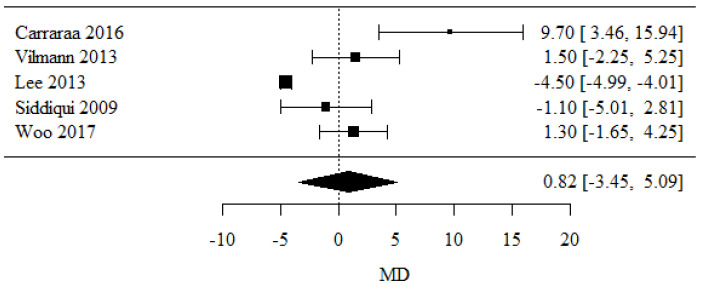
Mean size of lesions [25,26,27,28,29].

**Figure 8 cancers-16-02266-f008:**
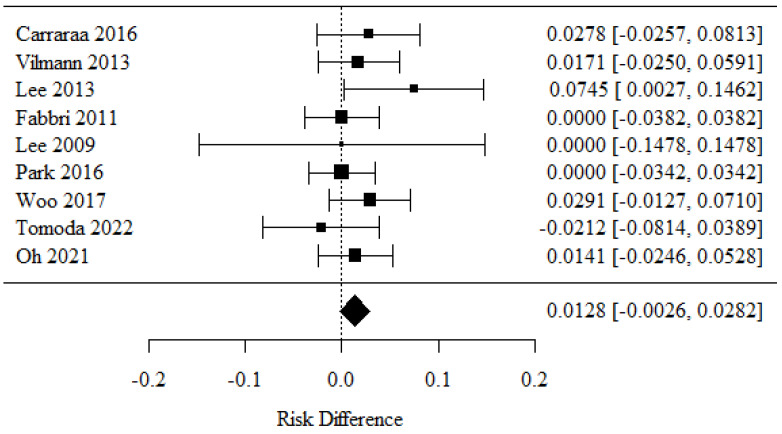
The rate of complications [26,27,28,29,30,31,33,34,36].

**Figure 9 cancers-16-02266-f009:**
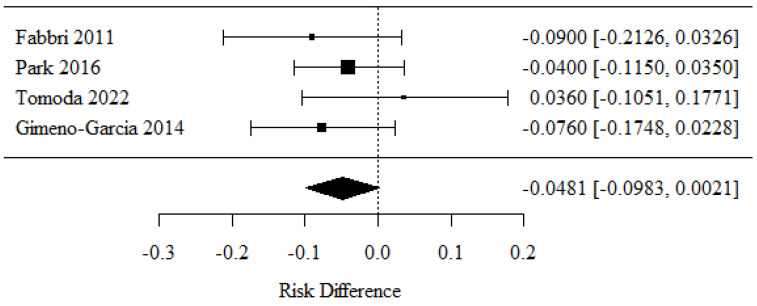
Sensitivity [30,34,35,36].

**Table 1 cancers-16-02266-t001:** Summary of the analyzed studies.

Author	Year	Number of Centers	Needle Passage through the Same Lesion	Number of Patients		Mean Age, Years	
				22	25	22	25
Silvia Carraraa [26]	2016	Single-center	No	72	72	66 ± 12	67 ± 12
Peter Vilmann [27]	2013	Multicenter	No	62	73	62 ± 13.6	64 ± 11.4
Jun Kyu Lee [28]	2013	Single-center	No	94	94	58.5 ± 11.8	61.3 ± 11.1
Uzma D. Siddiqui [25]	2009	Multicenter	No	64	67	69.3 ± 12.0	71.5 ± 12.0
Carlo Fabbri [36]	2011	Single-center	Yes	50		68.2 ± 7.4	
Hiroo Imazu [37]	2009	Single-center	Yes	43		NA	
Jeffrey H. Lee [33]	2009	Single-center	Yes	12		NA	
Georgios Mavrogenis [32]	2015	Single-center	Yes	28		69 (38–88)	
Se Woo Park [34]	2016	Single-center	Yes	56		65.8 ± 9.5 (44–89)	
Young Sik Woo [29]	2017	Single-center	No	103	103	61.2 ± 12.8	61.3 ± 11.6
Takeshi Tomoda [30]	2022	Single-center	Yes	43	45	72 (62–79)	70 (61–76)
Antonio Gimeno-García [35]	2014	Single-center	Yes	120		65.6 ± 11.3	
Dongwook Oh [31]	2021	Single-center	No	70	70	61.8 (9.27)	65.6 (9.15)
Hiroki Sakamoto [21]	2009	Single-center	Yes	24		NA	

## Data Availability

The data analyzed in this study were a reanalysis of existing data, which are openly available in the sources cited in the “References” section.

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
