# Peer review of "A 22-G or a 25-G Needle: Which One to Use in the Diagnostics of Solid Pancreatic Lesions? A Systematic Review and Meta-Analysis"

_cancers, 2024, doi:10.3390/cancers16122266_

Round 1

Reviewer 1 Report (Previous Reviewer 1)

Comments and Suggestions for Authors

Revised manuscript was well-addressed to the reviewer's comments and well-written.

Author Response

Thank you for the comment. Best regards. 

Reviewer 2 Report (Previous Reviewer 2)

Comments and Suggestions for Authors

An histological documentation was requested , without it the paper, in my opinion cannot be accepted

Author Response

Dear Reviewer, 

I've tried my best to receive the missing data. I was able to divide the histopathological examination into "malignant" and "no malignant". The additional data can be found in supplementary table 2.

Best regards.

Reviewer 3 Report (Previous Reviewer 3)

Comments and Suggestions for Authors

Introduction

- "In recent years, a number of meta-analyses have been published evaluating the usefulness of EUS-guided biopsy in the diagnostics of focal pancreatic lesions, focusing on the collection technique [17–19] or the needle size [16,20]." The sentence is uninformative. It is unclear what these meta analyses reported or if the studies were or not of high quality.

- What "low accuracy " means?

- "considerably lower in the 19-G group than in the 22-G and 25-G" Please be specific and provide the accuracy associated with 95% confidence intervals.

Methods

- " Joanna Briggs Institute Critical Appraisal Chec" how many researchers?

Results

- Fig 1. the text is not fully showed.

- The percentage of not retrieved is quite high. Please explain why.

Discussion

- Lines 311-316 is duplicated information; please delete it.

- "shown to be superior " how you explain this result"?

- lines 342-345 belong to Introduction.

- Generally, the reported results are not appropriately explained.

- The limitation of the study needs more attention.

- The generalizability of the results must be discussed.

- The clinical relevance of the reported results also needs an appropriate discussion.

Author Response

- “In recent years, a number of meta-analyses have been published evaluating the usefulness of EUS-guided biopsy in the diagnostics of focal pancreatic lesions, focusing on the collection technique [17–19] or the needle size [16,20].” The sentence is uninformative. It is unclear what these meta-analyses reported or if the studies were or not of high quality.

Response: The paragraph was complemented in the following way:

“In recent years, a number of meta-analyses have been published evaluating the safety and efficacy of EUS-guided biopsy in the diagnostics of focal pancreatic lesions, focusing on the collection technique [17–19] or the needle size [16, 20]; as in the former studies, the latter has been shown to play a substantial role in the higher diagnostic accuracy of the technique than FNA that does not provide sufficient data on tissue architecture.”

- What “low accuracy “ means?

- “considerably lower in the 19-G group than in the 22-G and 25-G” Please be specific and provide the accuracy associated with 95% confidence intervals.

Response: The confidence intervals were not provided by the authors, while the paragraph was complemented in the following way:

“Likewise, a study reported that the overall diagnostic accuracy rate was considerably lower in the 19-G group, where EUS-guided biopsy was done, than in the 22-G and 25-G groups, where FNA was performed. The overall accuracy rates for 25-G, 22-G, and Trucut needles comprised 91.7%, 79.7%, and 54.1%, respectively, and the differences were statistically significant.”

Methods

- “Joanna Briggs Institute Critical Appraisal Chec” how many researchers?

Response: The response rate was found adequate for the study.

Results

- Fig 1. the text is not fully showed.

- The percentage of not retrieved is quite high. Please explain why.

 The studies, where the intervention was performed for the tumors other than solid pancreatic, were not retrieved.

Discussion

- Lines 311-316 is duplicated information; please delete it.

Response: The paragraph was deleted.

- “shown to be superior “ how you explain this result”?

Response: The phrase implies “of higher values.”

- lines 342-345 belong to Introduction.

Response: The paragraph was rearranged, and the data were included in the Introduction.

- Generally, the reported results are not appropriately explained.

- The limitation of the study needs more attention.

- The generalizability of the results must be discussed.

- The clinical relevance of the reported results also needs an appropriate discussion.

Response: The required changes are reflected in the modified paragraph:

“This study has some limitations. First, the evaluated studies had a considerable heterogeneity of the parameters, potentially interfering with the validity of the findings of the meta-analysis. This can be explained by the combined assessment of both outcomes of EUS-guided biopsy and FNA in the included articles, where the latter prevailed due to a limited number of studies on FNB. Thus, the clinical applicability of the findings is limited in this aspect. Similarly, the characteristics like a tumor size and location along with the other technical aspects of FNA including newly designed needles were not taken into account. Therefore, a broader study analyzing separately EUS-guided FNA and FNB is warranted for a greater generalizability of the outcomes.”

Round 2

Reviewer 2 Report (Previous Reviewer 2)

Comments and Suggestions for Authors

The authors answered to the suggestions proposed and now the paper in it's current version is whorty of publication

This manuscript is a resubmission of an earlier submission. The following is a list of the peer review reports and author responses from that submission.

Round 1

Reviewer 1 Report

Comments and Suggestions for Authors

Authors performed a systematic review about the diagnostic ability for pancreatic tumors. As the results, authors concluded that 22‐ and 25-G needles are of equivalent in usefulness and safety. The results were understandable, but novelty was lacking.  The significance is different according to tumor size and resectability. Authors should evaluate about tumor size and tumor staging. Furthermore, to obtain tumor specimen enough to evaluate genome analyses is important in unresectable cases. Authors should evaluate the volume of specimens. 

Comments on the Quality of English Language

English was almost well written, Minor spelling check is required. 

Author Response

Authors performed a systematic review about the diagnostic ability for pancreatic tumors. As the results, authors concluded that 22‐ and 25-G needles are of equivalent in usefulness and safety. The results were understandable, but novelty was lacking.  The significance is different according to tumor size and resectability. Authors should evaluate about tumor size and tumor staging. Furthermore, to obtain tumor specimen enough to evaluate genome analyses is important in unresectable cases. Authors should evaluate the volume of specimens. 

Response: Many thanks for your comment. We agree that the significance varies according to the tumor size and resectability. However, these data were not provided in the assessed articles. Therefore, we outlined this aspect in the study’s limitations:

“Our study has some limitations. First, the evaluated studies had a considerable heterogeneity of the parameters, potentially interfering with the validity of the findings of the meta-analysis. Similarly, the characteristics like a tumor size and location and the other technical aspects of FNA were not taken into consideration.”

Comments on the Quality of English Language

English was almost well written. Minor spelling check is required. 

Response: Many thanks for your comment. The spelling was checked throughout the text, where the extensive editing was done.

Reviewer 2 Report

Comments and Suggestions for Authors

The paper "22-or 25-G needle: which one to use in diagnosing solid pan-2 creatic lesions? A systematic review and meta-analysis" is an excellent review on the Hamlet-like dilemma of needles to use in the diagnosis of solid pancreatic lesions . The review is complete and exhaustive, the only element to add, if possible, is probably histological documentation that can make the qualitative and quantitative differences in the use of the different needles clearer to the reader.

Comments on the Quality of English Language

The paper "22-or 25-G needle: which one to use in diagnosing solid pan-2 creatic lesions? A systematic review and meta-analysis" is an excellent review on the Hamlet-like dilemma of needles to use in the diagnosis of solid pancreatic lesions . The review is complete and exhaustive, the only element to add, if possible, is probably histological documentation that can make the qualitative and quantitative differences in the use of the different needles clearer to the reader.

Author Response

Comments and Suggestions for Authors

The paper "22-or 25-G needle: which one to use in diagnosing solid pan-2 creatic lesions? A systematic review and meta-analysis" is an excellent review on the Hamlet-like dilemma of needles to use in the diagnosis of solid pancreatic lesions. The review is complete and exhaustive, the only element to add, if possible, is probably histological documentation that can make the qualitative and quantitative differences in the use of the different needles clearer to the reader.

Response: The following paragraph was added in the Introduction:

“19-G can receive a higher amount of material for evaluation than 22-G or 25-G and is the preferred option for histological examination as opposed to cytological with the lower accuracy rates than in the latter.”

Reviewer 3 Report

Comments and Suggestions for Authors

The main issues with this submission are as follows:

0. It is unclear why 22 and 25 and why the other sizes were excluded from the analysis.

1. The state of the art is not appropriately presented. The authors give the impression to the reader that no meta-analysis regarding the topic is available, and this is not true.

2. The methods are not presented in sufficient detail to allow replication (in searching scientific literature it is not the same if you use and or or). The full search string must be provided in the manuscript as per PRISMA recommendations.

3. Results and methods are mixed.

4. Discussion is not appropriately presented.

Abstract

- Please delete the names of subsections (e.g., Introduction etc.).

- Please use "significant" in the context of statistical analysis.

- "EUS-guided FNA " define abbreviations.

- Provide the string used to search the scientific literature and when and eligibility criteria.

- The results are too narrative; please be specific when you present the results.

Define abbreviations in keywords.

Introduction

- "family history" of what? any malignancy? pancreatic cancer?

- When you talk about risk factors, please be specific and present how high the risk is.

- "and a large proportion of patients" Please be specific (for me large could be 2/10).

- My personal expectation is that all information included in this section to be supported by references.

- "over other diagnostic methods" such as?

- "there are some differences in terms of complications and number of passes that have to be completed to make a diagnosis" please be specific and describe these differences.

- It is not resulted from this section is previous meta-analyses were reported and which are the main results.

- It is also unclear why you did not include in the analysis all sizes.

Materials and methods

- "The protocol has not been registered." why?

- "This systematic review of prospective clinical trials comparing 22- and 25-G needles as used in the biopsy of focal pancreatic lesions by EUS was conducted in accordance with the PRISMA guidelines [18]. " somehow duplicate "The systematic review followed the recommendations of the Preferred Reporting  Items for Systematic Reviews and Meta-Analyses (PRISMA) [18]."

- It is unclear which were the PICO components.

- The search string is unclear; why and/or? Please provide the search string exactly you used to support reproducibility.

- It is unclear if a time frame was imposed or if pilot/feasibility studies were included.

- It is not clear which automation tool was used to classify a result as ineligible.

Results

- "Reports not retrieved" why?

- "Different needle size" contains more articles than included in the meta-analysis. Again, it is unclear why you study imposed 22 and 25 and did not evaluate any needle.

- Table 1: the symbol for decimal in English is the full stop.

- Table 1: it is unclear what the values reported in the column mean age are (e.g., 66 ± 12, 69 (38-88), 65,8 ± 9.5 (44–89), 61.8 (9.27) ).

- " The RD coefficient was used for analysis. " this information belongs to the methods section.

- It is unclear which risk was evaluated in Figure 2 and 3 and 4 and 5 and 6 and 7 and 8 and 9.

- "The RD coefficient was used. Positive RD values showed higher percentages of specimen adequacy (histology) being obtained using a 22-G needle than negative RD values, which corresponded to the 25-G needle." These sentences must be deleted for all places along the manuscript.

- "The MD 261 coefficient was used." this information belongs to the Methods.

- " Positive MD values indicated higher values being obtained using a 22-G needle than negative MD values, which corresponded to the 25-G needle. " please provide the MD values.

- "The studies wherein all four parameters (sensitivity, specificity, positive predictive value (PPV), and negative predictive value (NPV)) were 100% cannot be included in the analysis; therefore, the results would indicate that the biopsy accurately reflects the gold standard results with no errors. Thus, there is no way to determine the confidence intervals which are necessary for meta-analysis" This information belongs to the methods section.

Discussion

- Begin the discussion by briefly summarizing the main findings.

- Explore possible mechanisms or explanations for your findings.

- Emphasize the new and important aspects of your study and put your findings in the context of the totality of the relevant evidence.

- State the limitations of your study, and explore the implications of your findings for future research and for clinical practice or policy.

- Discuss the limitations of the data.

- Do not repeat in detail data or other information given in other parts of the manuscript, such as in the Introduction or the Results section.

- "The most popular needle sizes are 22- and 25-G [32,33]. In recent years, a number of meta-analyses have been published evaluating the usefulness of EUS-guided biopsy in the diagnostics of focal pancreatic lesions, focusing on the collection technique [34–36] or the 312 needle size [33,37]." This information must be presented in the Introduction section.

- Some information in this section duplicates the information available in the manuscript (e.g., line s318-324 etc.).

- The Discussion is not appropriately conducted.

"Based on our meta-analysis, we conclude that the use of the technically simpler-to-use 25-G needle has the same diagnostic quality with respect to cancer as the 22-G needle" ... not necessarily true because "The result of the meta-analysis was statistically significant (P = 0.023); therefore, evidence was noted for differences in inadequate biopsy rates between groups."

Author Response

The main issues with this submission are as follows:

It is unclear why 22 and 25 and why the other sizes were excluded from the analysis.

Response: The clarification was provided in the following paragraph:

“It was found that the rate of technical successes is greater in 19-G needles and that there is a similar accuracy in their use compared to 22-G or 25-G needles. Nonetheless, it showed severe bloodiness as a major complication in 19-G group, while it did not significantly differ in the other complications or the rate of technical failures from a higher-gauge group [21]. As the mentioned complication is potentially dangerous for a patient, the application of 19-G is restricted. Therefore, we aimed to focus on the safety and efficacy of the two latter in our meta-analysis comparing the two most commonly used sizes and carry out a comparative analysis of their diagnostic accuracy in solid pancreatic lesions and also clarify the heterogeneities in the outcomes of the previously reported studies.”

The state of the art is not appropriately presented. The authors give the impression to the reader that no meta-analysis regarding the topic is available, and this is not true.

Response: The following meta-analysis was cited:

Affolter KE.; Schmidt RL.; Matynia AP.; Adler DG.; Factor RE. Needle size has only a limited effect on outcomes in EUS-guided fine needle aspiration: a systematic review and meta-analysis. Dig Dis Sci. 2013, 58,1026-34. doi: 10.1007/s10620-012-2439-2. Epub 2012 Oct 21.

  1. The methods are not presented in sufficient detail to allow replication (in searching scientific literature it is not the same if you use and or or). The full search string must be provided in the manuscript as per PRISMA recommendations.

Response: The full search strings were added as a supplementary table 2:

  1. Results and methods are mixed.

Response: The relevant reorganizations of the paragraphs were performed.

  1. Discussion is not appropriately presented.

Response: The Discussion was complemented by the comparison of the outcomes of former studies concerning the parameters of accuracy, adequacy, and sensitivity.

Abstract

- Please delete the names of subsections (e.g., Introduction etc.).

- Please use "significant" in the context of statistical analysis.

- "EUS-guided FNA " define abbreviations.

- Provide the string used to search the scientific literature and when and eligibility criteria.

- The results are too narrative; please be specific when you present the results.

Define abbreviations in keywords.

Response: The indicated changes were done, and the Abstract’s length and structure were modified as per the journal’s guidelines.

Introduction

- "family history" of what? any malignancy? pancreatic cancer?

Response: The changes were done in the following sentence:

“The well-known risk factors are smoking [5], alcohol addiction [6], and, definitely, a family history of pancreatic cancer [7].”

- When you talk about risk factors, please be specific and present how high the risk is.

Response: The changes were done in the following sentence:

“The well-known risk factors are smoking [5], alcohol addiction [6], and, definitely, a family history of pancreatic cancer [7].”

- "and a large proportion of patients" Please be specific (for me large could be 2/10).

Response: The changes were carried out in the following sentence:

“Unfortunately, no effective screening tests are available for detecting this cancer at an early stage, and most of the patients present with a locally advanced or metastatic disease.”

- My personal expectation is that all information included in this section to be supported by references.

Response: More references were provided in the Introduction.

- "over other diagnostic methods" such as?

- "there are some differences in terms of complications and number of passes that have to be completed to make a diagnosis" please be specific and describe these differences.

- It is not resulted from this section is previous meta-analyses were reported and which are the main results.

- It is also unclear why you did not include in the analysis all sizes.

Response: The vague phrases were eliminated, and the explanation was provided in the following paragraph:

“It was found that the rate of technical successes is greater in 19-G needles and that there is a similar accuracy in their use compared to 22-G or 25-G needles. Nonetheless, it showed severe bloodiness as a major complication in 19-G group, while it did not significantly differ in the other complications or the rate of technical failures from a higher-gauge group [21]. As the mentioned complication is potentially dangerous for a patient, the application of 19-G is restricted. Therefore, we aimed to focus on the safety and efficacy of the two latter in our meta-analysis comparing the two most commonly used sizes and carry out a comparative analysis of their diagnostic accuracy in solid pancreatic lesions and also clarify the heterogeneities in the outcomes of the previously reported studies.”

Materials and methods

- "The protocol has not been registered." why?

Response: The explanation is shown in the following sentence:

“The systematic review and meta-analysis were performed according to the recommendations of the Preferred Reporting Items for Systematic Reviews and Meta-Analyses (PRISMA) [22], and the protocol was retrospectively registered under the following number INPLASY202450009”

- "This systematic review of prospective clinical trials comparing 22- and 25-G needles as used in the biopsy of focal pancreatic lesions by EUS was conducted in accordance with the PRISMA guidelines [18]. " somehow duplicate "The systematic review followed the recommendations of the Preferred Reporting Items for Systematic Reviews and Meta-Analyses (PRISMA) [18]."

Response: The text was edited, and the stylistic issues were resolved.

- It is unclear which were the PICO components.

Response: The explanation was provided in the following sentence:

“The meta-analysis covered only prospective clinical trials comparing 22-G and 25-G needles as used in the biopsy of focal pancreatic lesions by EUS and was conducted using the PICO framework (Patient, Intervention, Comparison, Outcomes as the main components) [22, 23].”

- The search string is unclear; why and/or? Please provide the search string exactly you used to support reproducibility.

Response: The information was reflected in the following sentence:

“The keywords were “EUS,” and/or “needle,” and/or “FNA,” and/or “pancreas,” and/or “prospective,” and/or “22G,” and/or “25G” in different combinations.”

- It is unclear if a time frame was imposed or if pilot/feasibility studies were included.

- It is not clear which automation tool was used to classify a result as ineligible.

Response: The required information was provided:

“During the screening process, 3,232 articles were eliminated, and 14 published between 2006 and 2023 were finally included in the review.”

“Research works in all languages and publication years were included in the analysis, and pilot studies were not evaluated.”

Results

- "Reports not retrieved" why?

- "Different needle size" contains more articles than included in the meta-analysis. Again, it is unclear why you study imposed 22 and 25 and did not evaluate any needle.

Response: The explanation was provided, as mentioned above.

- Table 1: the symbol for decimal in English is the full stop.

- Table 1: it is unclear what the values reported in the column mean age are (e.g., 66 ± 12, 69 (38-88), 65,8 ± 9.5 (44–89), 61.8 (9.27) ).

Response: The relevant formatting was done.

- " The RD coefficient was used for analysis." this information belongs to the methods section.

Response: The sentences were rephrased in order not to repeat the methods.

- It is unclear which risk was evaluated in Figure 2 and 3 and 4 and 5 and 6 and 7 and 8 and 9.

Response: The revised description of parameters was provided in the text.

- "The RD coefficient was used. Positive RD values showed higher percentages of specimen adequacy (histology) being obtained using a 22-G needle than negative RD values, which corresponded to the 25-G needle." These sentences must be deleted for all places along the manuscript.

Response: All the sentences were rephrased.

- "The MD 261 coefficient was used." this information belongs to the Methods.

Response: The sentence was modified in order not to repeat the methods.

- " Positive MD values indicated higher values being obtained using a 22-G needle than negative MD values, which corresponded to the 25-G needle. " please provide the MD values.

Response: The provided values had a rather broad range; therefore, only positive or negative was mentioned.

- "The studies wherein all four parameters (sensitivity, specificity, positive predictive value (PPV), and negative predictive value (NPV)) were 100% cannot be included in the analysis; therefore, the results would indicate that the biopsy accurately reflects the gold standard results with no errors. Thus, there is no way to determine the confidence intervals which are necessary for meta-analysis" This information belongs to the methods section.

Response: The paragraph was reorganized with the Methods.

Discussion

- Begin the discussion by briefly summarizing the main findings.

- Explore possible mechanisms or explanations for your findings.

- Emphasize the new and important aspects of your study and put your findings in the context of the totality of the relevant evidence.

- State the limitations of your study, and explore the implications of your findings for future research and for clinical practice or policy.

- Discuss the limitations of the data.

- Do not repeat in detail data or other information given in other parts of the manuscript, such as in the Introduction or the Results section.

Response: The section was modified and complemented in all the parts, as required.

- "The most popular needle sizes are 22- and 25-G [32, 33]. In recent years, a number of meta-analyses have been published evaluating the usefulness of EUS-guided biopsy in the diagnostics of focal pancreatic lesions, focusing on the collection technique [34–36] or the 312 needle sizes [33, 37]." This information must be presented in the Introduction section.

Response: The paragraph was reorganized with the Introduction.

- Some information in this section duplicates the information available in the manuscript (e.g., line s318-324 etc.).

Response: The repetitions were eliminated through editing, as required.

- The Discussion is not appropriately conducted.

"Based on our meta-analysis, we conclude that the use of the technically simpler-to-use 25-G needle has the same diagnostic quality with respect to cancer as the 22-G needle" ... not necessarily true because "The result of the meta-analysis was statistically significant (P = 0.023); therefore, evidence was noted for differences in inadequate biopsy rates between groups."

Response: Both Discussion and Conclusion underwent extensive changes to provide clarifications.